# ATLAS: BENCHMARKING AND ADAPTING LLMS FOR GLOBAL TRADE VIA HARMONIZED TARIFF CODE CLASSIFICATION

## ABSTRACT

Accurate classification under the Harmonized Tariff Schedule (HTS) is a critical yet underexplored problem in global trade compliance, where errors can delay shipments and disrupt supply chains. We present ATLAS, the first benchmark and fine-tuned large language model for HTS code prediction, constructed from the U.S. Customs Rulings Online Search System (CROSS). The benchmark includes 18,731 legally grounded rulings spanning 2,992 unique codes, reformatted into reasoning-oriented prompts. Our fine-tuned ATLAS model (LLaMA-3.3-70B) achieves $40\%$ accuracy at the full 10-digit level and $57.5\%$ at the 6-digit level—improvements of $+15$ and $+27.5$ points over strong baselines—while being approximately $5\times$ cheaper to deploy. These results establish HTS classification as a rigorous benchmark for hierarchical reasoning, cost-efficient adaptation, and alignment in domain-specialized large language models. The dataset and model are publicly released to encourage further research on structured reasoning for real-world compliance tasks.

[1]

## 1 INTRODUCTION

Every product imported into the global market must be assigned a Harmonized Tariff Schedule (HTS) code—a ten-digit identifier standardized by the World Customs Organization (WCO). The first six digits are harmonized globally, while the last four are country-specific; both are required for U.S. customs compliance Commission (2025).

The HTS is hierarchical: 22 sections expand into 99 chapters and thousands of subheadings, making tariff assignment a natural hierarchical learning problem. Six-digit accuracy reflects global consistency, while ten-digit accuracy measures U.S.-specific compliance.

Despite its centrality, classification remains a major bottleneck. The HTS spans over 17,000 pages, and recent U.S. policy changes mandate valid HTS codes for imports above $100. In 2025, postal operators such as India Post and Deutsche Post suspended deliveries to the U.S. due to missing or incorrect HTS codes tim (2025); reu (2025); usa (2025), illustrating how fragile trade becomes without scalable automation.

Large language models (LLMs) offer a scalable alternative: their semantic reasoning and structured prediction capabilities suit fine-grained distinctions (e.g., semiconductor wafers vs. finished chips). Moreover, since the first six digits are globally harmonized, improvements in HTS classification can generalize worldwide while directly addressing U.S. compliance needs.

### 1.1 CONTRIBUTIONS

We focus on the high-value semiconductor domain and present:

---

[1]1. HTS CROSS Rulings Dataset: `https://huggingface.co/datasets/flexifyai/cross_rulings_hts_dataset_for_tariffs`
2. Atlas LLM Model: `https://huggingface.co/flexifyai/atlas-llama3.3-70b-hts-classification`

- The first open-source benchmark for HTS classification Yuvraj & Devarakonda (2025a), derived from the U.S. Customs Rulings Online Search System (CROSS);

- A comprehensive evaluation of leading proprietary and open-source models, including GPT-5-Thinking, Gemini-2.5-Pro-Thinking, LLaMA-3.3-70B, DeepSeek-R1, and GPT-OSS-120B;

- The fine-tuned ATLAS model Yuvraj & Devarakonda (2025b) (LLaMA-3.3-70B), achieving $40\%$ 10-digit and $57.5\%$ 6-digit accuracy—substantially outperforming baselines—while being up to $8\times$ cheaper and fully self-hostable for privacy-sensitive deployments.

Together, these contributions position tariff code classification as a new benchmark for evaluating reasoning and adaptation in large language models.

## 2  DATASET

Our main contribution is the first large-scale dataset for Harmonized Tariff Schedule (HTS) classification, derived from the U.S. Customs Rulings Online Search System (CROSS) Customs & Protection (2025). CROSS contains legally binding rulings by U.S. Customs and Border Protection (CBP) specifying the correct 10-digit HTS code for products. These rulings are authoritative yet dispersed across thousands of HTML pages, previously inaccessible for ML research.

### 2.1  COLLECTION AND SCOPE

We built an automated agent Project (2025); Google (2025); Pirogov (2025) to scrape CROSS and align each ruling with its official 10-digit HTS code from Commission (2025). Focusing on semiconductor and manufacturing chapters, we obtained $18{,}731$ rulings covering $2{,}992$ unique codes. Frequent rulings highlight ambiguous or high-demand categories, while absent codes suggest stable ones. Table 1 lists representative chapters (the complete distribution is provided in Appendix A, Table 5).

Table 1: Sample distribution of CROSS rulings across major HTS chapters.

| Chapter | Codes | Rulings | Description |
|---|---|---|---|
| 39 | 264 | 2781 | Plastics and articles thereof |
| 61 | 163 | 3445 | Apparel, knitted or crocheted |
| 73 | 389 | 1749 | Articles of iron or steel |
| 84 | 801 | 3566 | Machinery and mechanical appliances |
| 85 | 293 | 1445 | Electrical machinery and equipment |
| 90 | 256 | 1499 | Optical and precision instruments |
| **All** | 2992 | 18731 | Combined total |

### 2.2  TRANSFORMATION TO MODEL-READY FORMAT

Raw rulings are verbose legal letters. We used GPT-4o-mini OpenAI et al. (2024) for information extraction, converting each into a concise instruction–response pair containing (a) a product description, (b) reasoning trace, and (c) final HTS code. The complete prompt template is provided inAppendix B. This structure enforces reasoning-based prediction, aligning with chain-of-thought research Wei et al. (2023).

### 2.3  SPLITS AND AVAILABILITY

We reserved 200 samples each for validation and testing, with the remaining $18{,}254$ for training (Table 2). To ensure fair and representative evaluation despite the small test size, the 200 test samples were stratified across high-variance HTS chapters (e.g., 84, 85, and 90) to reflect the diversity and ambiguity observed in real-world tariff rulings. The dataset is publicly available on Hugging Face Yuvraj & Devarakonda (2025a).

Table 2: CROSS dataset splits.

| Training | 18,254 |
|---|---|
| Validation | 200 |
| Test | 200 |

## 2.4 DISCUSSION

HTS rulings demand fine-grained reasoning (e.g., partially vs. fully fabricated wafers) and hierarchical accuracy at 6- and 10-digit levels. Errors have direct compliance costs, making this dataset a realistic and impactful benchmark for evaluating structured reasoning in large language models.

## 3 MODEL TRAINING

While several open-source large language models could, in principle, be adapted for tariff classification, we made a deliberate and principled choice to focus exclusively on **LLaMA-3.3-70B** Grattafiori et al. (2024). Two factors motivated this decision. First, practical *budget constraints* made it infeasible to fine-tune multiple frontier models at scale. Second, LLaMA-3.3-70B is a dense architecture, making it both simpler to fine-tune and easier to deploy in inference settings compared to Mixture-of-Experts (MoE) architectures such as DeepSeek-R1 or GPT-OSS-120B. From a community perspective, providing a dense and reproducible baseline lowers the entry barrier for downstream research: training and inference pipelines are easier to set up, memory usage is more predictable, and accuracy is less sensitive to expert routing heuristics.

### 3.1 SUPERVISED FINE-TUNING OBJECTIVE

We adapted LLaMA-3.3-70B to the CROSS dataset using supervised fine-tuning (SFT) Brown et al. (2020); Ouyang et al. (2022). Each ruling was transformed into an input–output pair, where the input is a ruling-derived product description and the output is the correct HTS code along with a reasoning trace. This makes the task well aligned with the SFT paradigm, which minimizes the token-level negative log-likelihood of ground-truth outputs.

Formally, for an input sequence $x = (x_1, \ldots, x_n)$ and target sequence $y = (y_1, \ldots, y_m)$, the model with parameters $\theta$ defines conditional probabilities $p_\theta(y_t \mid x, y_{<t})$. The training loss is then:

$$\mathcal{L}_{\text{SFT}}(\theta) = -\sum_{t=1}^{m} \log p_\theta(y_t \mid x, y_{<t}),$$

which corresponds to the standard negative log-likelihood objective.

### 3.2 TRAINING SETUP AND STABILITY

Fine-tuning was performed for 5 epochs (approximately 1,400 steps) using the AdamW optimizer with $\beta_1 = 0.9$, $\beta_2 = 0.95$, weight decay $= 0.1$, and a cosine learning-rate schedule initialized at $1 \times 10^{-7}$. To manage the high memory footprint of 70B-parameter models, we employed bf16 precision and gradient accumulation to simulate a batch size of 64 sequences. Training was distributed across $16 \times$ A100-80GB GPUs using fully sharded data parallelism.

As shown in Figure 1, the training loss decreases sharply in the first 200 steps and then stabilizes near convergence, with no sign of overfitting on the validation set. We observed stable gradient norms and no catastrophic spikes in loss, suggesting that dense models like LLaMA-3.3-70B are well suited to small but domain-specific datasets when carefully regularized. This highlights that reproducible fine-tuning of frontier models is feasible even under modest compute budgets, provided that optimization choices are tuned to stability.

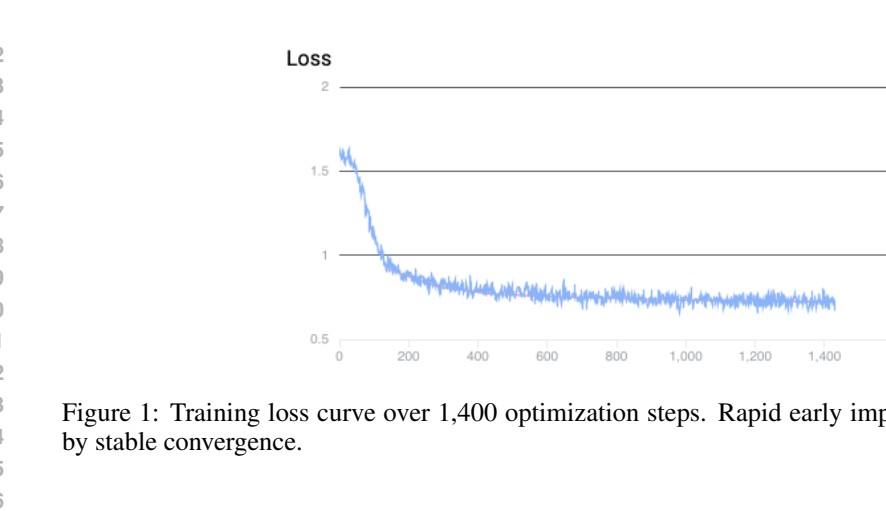

Figure 1: Training loss curve over 1,400 optimization steps. Rapid early improvement is followed by stable convergence.

### 3.3 BEYOND SUPERVISED FINE-TUNING: REINFORCEMENT LEARNING

Although SFT provides a strong domain-adapted baseline, it is inherently limited by the distribution of observed CROSS rulings. Reinforcement learning (RL) offers a pathway to improve generalization beyond exact supervision.

A lightweight and cost-effective starting point is a **rule-based reward model**. For instance, we can define rewards as: 1 when the model correctly predicts the full 10-digit HTS US code, 0.6 when the first 6 digits (globally harmonized HS code) are correct, and $-1$ otherwise. Formally, for classification $\hat{y}$ and gold label $y$:

$$R(\hat{y}, y) = \begin{cases} 1, & \text{if } \hat{y}_{1:10} = y_{1:10} \\ 0.6, & \text{if } \hat{y}_{1:6} = y_{1:6} \\ -1, & \text{otherwise.} \end{cases}$$

Such a structured reward can be readily integrated into GRPO Shao et al. (2024) or related policy-gradient methods such as PPO Schulman et al. (2017). This approach would allow the model to explore reasoning trajectories that go beyond memorization, while keeping the reward landscape interpretable and inexpensive to compute. Importantly, this positions tariff code classification as a promising candidate for lightweight reinforcement learning research on high-stakes, domain-specific reasoning tasks.

### 3.4 ABLATIONS AND FUTURE WORK

While our study focused exclusively on LLaMA-3.3-70B, several ablation studies could provide deeper insights and further guide the community:

- **Model scale:** Evaluating smaller LLaMA variants (e.g., 8B or 3B) would clarify the trade-off between accuracy, cost, and deployability on edge devices.

- **Retrieval augmentation:** Integrating retrieval over the 17,000-page HTS documents may reduce hallucinations and improve long-tail classification accuracy, complementing SFT.

- **Contrastive and hybrid objectives:** Beyond NLL, contrastive learning between closely related codes (e.g., semiconductor wafers vs. finished chips) may sharpen decision boundaries.

- **Direct Preference optimization:** Beyond NLL training, methods such as Direct Preference Optimization (DPO) Rafailov et al. (2023) could leverage structured preferences over HTS classifications (e.g., preferring correct 10-digit codes over near-misses, or valid reasoning traces over hallucinated ones). This would allow the model to learn not just to imitate CROSS rulings but to actively steer away from incorrect classifications.

- **RL scaling studies:** Comparing rule-based GRPO with preference-based RLHF could quantify the cost–benefit tradeoffs of reinforcement learning at 70B scale.

These directions highlight that while ATLAS establishes a strong dense-model baseline, HTS classification remains an open problem with substantial room for methodological innovation.

# 4 RESULTS AND EVALUATION

We evaluate all models on a held-out test set of 200 CROSS rulings, predicting the correct 10-digit HTS US code per product. Because the classification is hierarchical, we report three metrics: (1) full 10-digit match, (2) partial 6-digit match (globally harmonized), and (3) average digits correct (0–10).

## 4.1 ACCURACY AT 10 AND 6 DIGITS

Table 3 shows fully correct classifications. GPT-5-Thinking[2] achieves 25%, while **Atlas** attains **40%**, the highest among all models.. At the 6-digit level (Table 3), Atlas also leads with **57.5%**, slightly above GPT-5's 55.5%. These results confirm that domain-specific fine-tuning improves both global and U.S.-specific accuracy.

| Model | 10-digit (%) | 6-digit (%) | Avg. Digits |
|---|---|---|---|
| GPT-5-Thinking | 25.0 | 55.5 | 5.61 |
| Gemini-2.5-Pro-Thinking | 13.5 | 31.0 | 2.92 |
| DeepSeek-R1 (05/28) | 2.5 | 26.5 | 3.24 |
| GPT-OSS-120B | 1.5 | 8.0 | 2.58 |
| LLaMA-3.3-70B | 2.1 | 20.7 | 3.31 |
| **Atlas (fine-tuned)** | **40.0** | **57.5** | **6.30** |

Table 3: Model accuracy across 10-digit, 6-digit, and average-digit metrics.

Figure 2 visualizes the relative advantage of Atlas, particularly for 10-digit U.S.-specific codes.

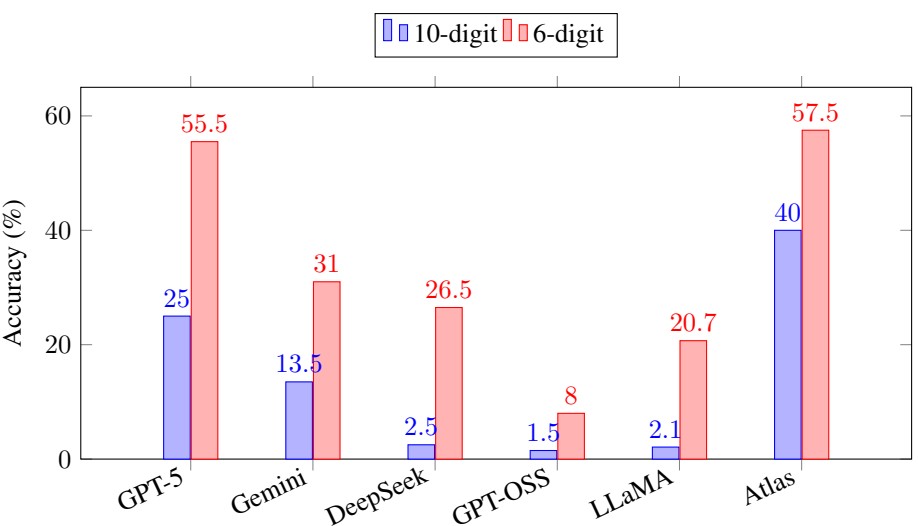

Figure 2: Comparison of model accuracy at 10- and 6-digit levels.

## 4.2 INFERENCE COST

Cost per classification is crucial for scalability. Table 4 shows that open-source models, particularly Atlas, are an order of magnitude cheaper while maintaining state-of-the-art accuracy.

---

[2]Model outputs and pricing were obtained from public API documentation and experiments conducted in September 2025.

| Model | Cost per 1,000 inferences (USD) |
|---|---|
| GPT-5-Thinking | 3.30 |
| Gemini-2.5-Pro-Thinking | 5.50 |
| DeepSeek-R1 | 1.00 |
| GPT-OSS-120B | 0.90 |
| LLaMA-3.3-70B | 0.70 |
| **Atlas (fine-tuned)** | **0.70** |

Table 4: Estimated cost for 1,000 HTS inferences.

### 4.3 DISCUSSION

Taken together, these results highlight a critical tradeoff: **Atlas** not only surpasses GPT-5-Thinking in accuracy (40% vs. 25% fully correct classifications), but also reduces inference cost by nearly **5× compared to GPT-5** and almost **8× compared to Gemini-2.5-Pro-Thinking**. Moreover, the strong performance on partially correct classifications demonstrates that Atlas generalizes beyond U.S.-specific tariffs to the globally harmonized 6-digit regime, reinforcing its utility for international trade applications. A qualitative comparison illustrating model reasoning differences is provided in Appendix C.

## 5 SUMMARY AND FUTURE DIRECTIONS

We introduced the first benchmark for Harmonized Tariff Schedule (HTS) code classification and presented ATLAS, a fine-tuned LLaMA-3.3-70B model for global trade compliance. The study establishes HTS classification as a challenging new LLM benchmark, with three main takeaways:

- **Performance:** ATLAS achieves 40% fully correct and 57.5% partially correct (6-digit) classifications, surpassing GPT-5-Thinking (+15 pts) and Gemini-2.5-Pro (+27.5 pts).
- **Efficiency:** ATLAS is **5× cheaper than GPT-5** and **8× cheaper than Gemini**, while supporting secure self-hosted deployment.
- **Challenge:** Even the best model attains only 40% 10-digit accuracy, underscoring substantial headroom for progress.

Future work includes expanding coverage beyond semiconductors, distilling Atlas into smaller (8B–3B) variants for edge use, and applying reinforcement learning via rule-based rewards Shao et al. (2024); Rafailov et al. (2023) to improve reasoning and alignment.

We release ATLAS Yuvraj & Devarakonda (2025b) and the dataset Yuvraj & Devarakonda (2025a) to encourage research on domain-specialized, trustworthy LLMs for global trade.

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

## A   FULL DATASET DISTRIBUTION

Table 5 lists the complete distribution of CBP rulings across all included Harmonized Tariff Schedule (HTS) chapters. This extended version complements the abbreviated table in Section 2.

Table 5: Full distribution of CBP rulings across HTS chapters.

| Chapter | HTS Codes | Rulings | Description |
|---|---|---|---|
| 27 | 3 | 7 | Mineral fuels, mineral oils and products of their distillation |
| 28 | 55 | 253 | Inorganic chemicals; organic or inorganic compounds of precious metals |
| 29 | 324 | 1199 | Organic chemicals |
| 32 | 16 | 44 | Tanning or dyeing extracts; dyes, pigments, tannins and derivatives |
| 35 | 23 | 306 | Albuminoidal substances; modified starches; glues; enzymes |
| 39 | 264 | 2781 | Plastics and articles thereof |
| 40 | 89 | 417 | Rubber and articles thereof |
| 42 | 28 | 223 | Articles of leather; saddlery, harness, travel goods, handbags |
| 49 | 1 | 12 | Printed books, newspapers, pictures, and other printed products |
| 61 | 163 | 3445 | Apparel and clothing accessories, knitted or crocheted |
| 70 | 17 | 36 | Glass and glassware |
| 72 | 10 | 15 | Iron and steel |
| 73 | 389 | 1749 | Articles of iron or steel |
| 74 | 52 | 119 | Copper and articles thereof |
| 76 | 27 | 80 | Aluminum and articles thereof |
| 82 | 152 | 1450 | Tools, implements, cutlery, spoons, and forks of base metal |
| 83 | 6 | 17 | Miscellaneous articles of base metal |
| 84 | 801 | 3566 | Nuclear reactors, boilers, machinery, and mechanical appliances |
| 85 | 293 | 1445 | Electrical machinery and equipment; sound recorders and reproducers |
| 87 | 17 | 60 | Vehicles (other than railway/tramway rolling stock) and parts |
| 90 | 256 | 1499 | Optical, photographic, measuring, precision, and medical instruments |
| 94 | 6 | 8 | Furniture; bedding, mattresses, cushions, and similar furnishings |
| **All** | 2992 | 18731 | All chapters combined |

## B  PROMPT TEMPLATE FOR DATA TRANSFORMATION

Each ruling was converted into a structured instruction–response pair to facilitate supervised fine-tuning of language models. The complete transformation prompt is shown below.

```
Given the following HTS ruling information:

HTS Code: {hts_code}
Ruling Number: {ruling_number}
Title: {title}
Date: {date}
URL: {url}
Summary: {summary}
Content: {content}

Please analyze this information and provide:

a) A concise product description representing the item being classified
b) A reasoning path justifying why the HTS US code is correct
c) The final HTS US code

Format your response as follows:

User: What is the HTS US Code for [product_description]?
Model:
HTS US Code -> [HTS US Code]
Reasoning -> [detailed_reasoning_path]
```

This design enforces explicit reasoning traces, aligning with recent advances in chain-of-thought modeling Wei et al. (2023).

## C  QUALITATIVE EXAMPLE

Table 6 presents one representative example from the CROSS test set, showing how ATLAS reasons through a real-world tariff classification case compared to other models. The excerpt highlights how Atlas distinguishes manufacturing stages and correctly identifies the applicable HTS code.

Table 6: Representative qualitative example from the CROSS dataset.

| | |
|---|---|
| **Product Description** | Silicon wafers, partially processed, used for semiconductor fabrication. |
| **Ground Truth HTS Code** | 3818.00.00.00 — Chemical elements doped for use in electronics. |
| **Atlas (Ours)** | *Reasoning:* Identifies item as doped wafer, not a finished semiconductor device; recognizes manufacturing stage. *Prediction:* 3818.00.00.00 (✓) Fully correct. |
| **GPT-5-Thinking** | *Reasoning:* Focuses on "semiconductor fabrication," misclassifies as complete integrated circuit. *Prediction:* 8542.31.00.00 (×) Incorrect; classifies as final chip. |
| **Gemini-2.5-Pro-Thinking** | *Reasoning:* Associates with silicon materials but ignores doping context. *Prediction:* 3824.99.99.99 (×) Incorrect; generic chemical compound. |

