# OpenReview forum: "ATLAS: Benchmarking and Adapting Large Language Models for Global Trade via Harmonized Tariff Code Classification"
_NeurIPS.cc/2025/Workshop_Mexico_City/NORA — NeurIPS 2025 Workshop NORA Poster_

### Official Review · Reviewer_9qme · 2025-11-01
**Clearly written paper on a new dataset**

**Rating:** 4
**Confidence:** 3

**Review:**

This is a clearly written paper reporting the fine-tuning of Llama-3.3 70B for predicting Harmonized Tariff Schedule (HTS) codes. The dataset was scraped online and made publicly available. The fine-tuned model was compared against several LLMs, including GPT-5-Thinking and Gemini-2.5-Pro-Thinking, and it significantly outperformed the other LLMs.

Strengths:
- The dataset is newly prepared and made publicly available.
- The paper is clearly written, and the contents are well organized.

Weaknesses:
- The focus of the paper is less relevant to “Knowledge Graphs & Agentic Systems”

In summary, this is a clearly written paper introducing a new dataset along with an LLM well fine-tuned to it. However, the focus of the paper may not be suitable for the current workshop and would be better directed toward a more appropriate venue.

---

### Official Review · Reviewer_spnx · 2025-11-02
**Review for the paper titled "ATLAS: Benchmarking and Adapting Large Language Models for Global Trade via Harmonized Tariff Code Classification" submitted to NORA workshop**

**Rating:** 6
**Confidence:** 3

**Review:**

Summary: The paper describes a benchmark for classifying the Harmonized Tariff Schedule (HTS) of the United States. The authors describe how the dataset was collected from the U.S. Customs Rulings Online Search System (CROSS), clean the data and transform it into pairs of product description and 10-digit HTS code. The authors then use supervise learning to fine-tune LLaMA-3.3-70B with reinforcement learning. Finally, the authors compare their model against several leading large language models, such as Deepseek or GPT5.

Strengths: + The paper is well written and simple to follow.

Weaknesses: - Unclear impact of the classification task beyond the US customs.
- Unclear relation to the workshop themes.
- Unclear how reinforcement learning is integrated in the model.
- Limited experimentation and discussion. The nicely-outlined ablation studies are not part of this paper.

Suggestion for improvement:
* Add an example right in the introduction to inform the reader about the task.
* Add an example of the dataset instances.
* Include details on how the quality of the dataset (mainly constructed through GPT4o-mini) was ensured.
* Briefly explain (maybe in a footnote) what "semiconductor domain" means.
* Explain the hierarchical classification you are referring to in the paper.
* Define what GRPO and RPO stand for and explain them briefly.
* Explain the integration of reinforcement learning in the fine-tuning.
* Define what NLL stands for

Minor:
The workshop uses a single-blind review process, yet the submission lists the authors as anonymous while including self-citations.

---

### Official Review · Reviewer_iFWd · 2025-11-03
**Quality Work but a Bit Overkill and Undermotivated**

**Rating:** 5
**Confidence:** 3

**Review:**

This paper describes the construction of a classification model designed to supply US Harmonized Tariff Codes to semiconductor goods. The authors scrape data from the US Customs website that contains rulings that are then used to fine-tune an LLM with cost considerations. The authors demonstrate that this low-cost fine-tuned LLM performs better than out-of-the-box LLMs that are more expensive.

The work itself seems fine but I don't know why the authors insist on using an LLM for this. They provide no prior or related work to motivate the use of LLMs here. I'm not convinced they couldn't extract features and train a Random Forest or XGBoost model and get higher accuracy than their LLM-based method. I'm also not clear how the current approach exploits the hierarchical nature of the data.

Furthermore, it's hard to get a sense of the model's performance. Are there certain subsets/portions of the hierarchy that are dominating this data set and/or the model is performing the best on those? It would be worth including this breakdown in the appendix. Also, using metrics other than accuracy would be helpful here.

The paper itself is clear enough to read except for citations.
1. Please place inline citations in parentheses, like this: "(Commission, 2025)". Reading a sentence like "both are required for U.S customs compliance Commission" is very jarring and confusing.
2. Please provide the names of the authors for documents like "Commission" and the news articles you cite. For example, your Reuters article shouldn't be "reu" but rather "Kaeckenhoff et al." with all names in the reference and the "venue" as Reuters. If it's an agency such as the US International Trade Commission, this should then say, "US International Trace Commission" and not just "Commission". (You can also abbreviate this once and continue to say "US ITC" throughout.)
3. Please do not cite the contributions of this paper if they have already been released since this removes the anonymity requirement of this reviewing process. Furthermore, the references in the bulleted list in section 1 should be footnotes with links. Saying you're providing "the first open-source benchmark" then citing to work that already exists is confusing.
4. Please try to maintain proper capitalization in your citations using `{ }` (e.g. "DHL", "German").

Unrelated to citations, please combine your future work sections into one section at the end. And please capitalize "Large Language Models" as such.

---

### Official Review · Reviewer_xvgU · 2025-11-05
**This paper introduces ATLAS, the first benchmark and fine-tuned large language model for Harmonized Tariff Schedule (HTS) code classification, demonstrating significant improvements in accuracy and cost-efficiency over existing models, and establishing HTS classification as a rigorous new task for evaluating structured reasoning in LLMs.**

**Rating:** 9
**Confidence:** 5

**Review:**

The paper presents a technically rigorous and impactful contribution to the field of domain-specific large language models (LLMs) through the introduction of ATLAS, a benchmark and fine-tuned model for Harmonised Tariff Schedule (HTS) classification. The authors tackle a real-world problem of global trade compliance, where accurate HTS code assignment is essential yet remains a bottleneck due to its complexity and hierarchical structure.

The quality of the work is high. The authors construct a legally grounded dataset from the U.S. Customs Rulings Online Search System (CROSS), comprising nearly 19,000 rulings across almost 3,000 unique HTS codes. They apply supervised fine-tuning to a dense LLaMA-3.3-70B model and demonstrate stable training, reproducibility and strong performance. The evaluation is thorough, comparing ATLAS against leading proprietary and open-source models using clear metrics (10-digit, 6-digit and average-digit accuracy), and showing substantial improvements in both accuracy and cost-efficiency.

Clarity is another strength of the paper. The writing is well-structured and accessible, with clear motivation, detailed methodology and informative tables and figures. The inclusion of qualitative examples and appendices enhances transparency and interpretability.
The originality of the work is notable. It introduces the first benchmark for HTS classification and reframes the task as a hierarchical reasoning challenge, aligning with structured prediction and chain-of-thought prompting. This is a novel and timely application of LLMs to a high-stakes domain that has not previously been explored in the machine learning literature.
The significance of the contribution is substantial. HTS classification affects international logistics and compliance, and the ATLAS model achieves state-of-the-art performance while being significantly more cost-effective and deployable than existing models. The public release of both the dataset and model supports reproducibility and encourages further research in trustworthy, domain-specialized LLMs.
The paper’s strengths include its novel benchmark and dataset, strong empirical results, cost-efficiency and relevance to a real-world problem with global impact. It is technically sound, clearly presented and methodologically innovative. While the domain scope is currently limited to semiconductors and manufacturing, and the test set is relatively small, these limitations do not detract significantly from the overall contribution.

In conclusion, this paper stands out for its originality, practical relevance and methodological rigour. It makes a meaningful advance in applying LLMs to structured reasoning tasks in compliance-critical domains.

---

### Official Review · Reviewer_qxF9 · 2025-11-06
**Interesting Task and Valuable Results, but Further Experiments and Analysis Are Needed**

**Rating:** 5
**Confidence:** 4

**Review:**

Summary:

This paper addresses a classification task under the Harmonized Tariff Schedule (HTS), a key challenge in global trade compliance.
The authors propose ATLAS, a model fine-tuned from LLaMA-3.3-70B, which achieves 40% accuracy at the 10-digit level and 57.5% at the 6-digit level—significantly outperforming strong baselines while being approximately five times more cost-efficient to deploy.

Strengths:

This paper makes a commendable contribution by tackling the HTS classification task, a critical and under-explored problem in global trade compliance. In addition to releasing a valuable dataset, the authors demonstrate measurable accuracy improvements through fine-tuning, achieving performance that surpasses several commercial baseline systems. These results highlight the practical significance of the proposed dataset and its potential to advance both academic and industrial applications in this domain.

Weaknesses and Areas for Improvement:

While the experimental results are encouraging, the methodological novelty is limited. The study primarily focuses on fine-tuning LLaMA-3.3-70B using the constructed dataset, without exploring alternative or innovative adaptation strategies. Consequently, the paper’s contribution lies mainly in performance validation rather than methodological advancement.

Specific Comments and Suggestions:

- Model size: It is unclear whether LLaMA-3.3-70B represents the optimal model choice. Comparative experiments with smaller models would help evaluate the trade-off between model size and performance.

- Training data sufficiency: An ablation study examining how training data size affects classification accuracy would provide valuable insight into data efficiency.

- Model generalization: To assess the generality of the findings, it would be beneficial to include results from other fine-tunable LLMs such as Mistral or Qwen, beyond the LLaMA family.

- Impact of reasoning models: Please clarify the performance difference when using reasoning models versus standard models, as this would help isolate the contribution of reasoning capabilities to overall accuracy.

- Performance comparison: The reported 6-digit accuracy of GPT-5-Thinking is comparable to that of ATLAS. A deeper analysis explaining the reasons behind this similarity would enhance interpretability.

- Paper organization: The discussion of future work currently appears in Section 3.4. Relocating this content to the concluding section would improve the paper’s logical flow and overall structure.

---

### Official Review · Reviewer_4WHB · 2025-11-06
**Interesting work but content doesn't match CFP**

**Rating:** 3
**Confidence:** 5

**Review:**

The authors address a challenging problem and space (very high-dimensional and nuanced classification problem in assigning tariff codes). They use supervised fine tuning to train an LLM (llama) and compare it to other approaches, achieving what appear to be reasonable results. However, the paper only covers the training and there is nothing relevant to the CFP. In particular there is no mention of knowledge graphs or agents/agentic systems, and though the authors mention reinforcement learning, it doesn't appear they performed any. With a deployment it might be a good fit for a *ACL industry track paper, but doesn't fit the brief for this workshop.

---

### Official Review · Reviewer_s3AK · 2025-11-07
**Application of SFT in a consequential domain**

**Rating:** 6
**Confidence:** 4

**Review:**

This paper is concise, easy to read and understand. Like many papers before it, a foundation model has been fine tuned in a supervised manner using domain-specific data and evaluation results are shared to demonstrate the utility/promise of the resulting model in the domain. However, the authors have picked a  real-world domain where the problem being solved suits the principles of SFT very well, so if developed properly, in future, such a model can have a significant impact on a consequential domain.

Some feedback/questions to the authors:
1. Hyperlink names could be better - section 2.1 first sentence - e.g. 'Project' 'Google' etc are not self-explanatory.
2. Why only 200 samples for validation and test? Asked another way, why did only 1.25% of the overall set get reserved for testing and/or why did 97.5% of the set get used for training given this is SFT? (e.g. concerns of overfitting)
3. Remark about ‘practical budget constraints’ could be evidenced further with some analysis on costing concerns considered in this paper's work.
4. SFT’s training loss stated as negative log-likelihood is sufficient; an equation is not needed (for brevity)
5. How does one interpret the significance of the ‘Avg Digit’ metric? How is it measured? Also, in Appendix where reasoning is compared, Gemini is close edit-distance wise to Atlas in the eventual classification. Was this considered as an area of comparison?
6. GPT5-Thinking without SFT does almost as well as Atlas with SFT on 6-digit accuracy. This is worth noting and discussing in the paper — esp section 4.3
7. It is more apt to say Atlas is an open-weight model (vs open-source).
8. Does Table 4 correspond to cost estimates from a certain inference provider? It would be good to explain how this measure was estimated - e.g. were multiple providers were surveyed?
9. Future work/direction is discussed in two sections 3.4 and 5. For coherence, it is preferable to consolidate them in one single section. It is okay to discuss whether a given future work item/direction corresponds to ablation vs something else within this section.
10. I think the greater contribution is the benchmark/dataset - deserves more discussion in the paper than it has received.

While this is a decent paper and aims to highlight real-world applicability of LLMs, its subject does not relate to the interplay between AI Agents and Knowledge graphs/systems, which is the primary purpose of this workshop. I have therefore proposed a marginal accept - ACs/PC please appraise accordingly.